# Are Post-Care Recommendations Following Upper-Face Botulinum Toxin Treatment Scientifically Necessary? A Retrospective Study Based on 5000 Patients

**DOI:** 10.3390/toxins17080372

**Published:** 2025-07-28

**Authors:** Adriano Santorelli, Giovanni Salti, Maurizio Cavallini, Salvatore Piero Fundarò, Matteo Basso, Martina Ponzo, Stefano Avvedimento, Stefano Uderzo

**Affiliations:** 1“Santorelli and Partners”, Via Raffaele Morghen, 88, 80129 Napoli, NA, Italy; 2Independent Researcher, Via Claudio Monteverdi, 2, 50144 Firenze, FI, Italy; 3Agorà, Via dei Ciclamini, 23, 20147 Milano, MI, Italy; 4Independent Researcher, Viale Schiocchi, 40, 41124 Modena, MO, Italy; 5Independent Researcher, Via Galata, 37/4, 16121 Genova, GE, Italy; 6Residency Program in Plastic, Reconstructive and Aesthetic Surgery, Campus Biomedico University, Via Álvaro del Portillo, 21, 00128 Roma, RM, Italy; 7“Ospedale Cardarelli”, Via Antonio Cardarelli, 9, 80131 Napoli, NA, Italy

**Keywords:** botox, aesthetic medicine, eyelid ptosis, botulinum toxin type A, injection protocol, forehead wrinkles

## Abstract

Background: Patient care following botulinum toxin injections has long been guided by anecdotal instructions, often based on theoretical considerations. This study evaluates the necessity of extended post-treatment instructions by analyzing outcomes and satisfaction in patients who followed only a 10 min precaution protocol. Materials and Methods: A retrospective, multicentric study was conducted across six Italian centers, analyzing 5014 patients treated with botulinum toxin for upper facial wrinkles between 2015 and 2020. Outcomes included adverse effects—particularly upper eyelid ptosis—and patient satisfaction. Follow-up was performed at two weeks. Results: No cases of upper eyelid ptosis were observed. Among 4000 patients who attended follow-up, adverse effects occurred in only 5.99%, notably lower than rates reported in the literature. Of the 2010 patients who completed the satisfaction questionnaire, 90% reported being very satisfied. These findings support the safety of limiting post-treatment instructions to 10 min. Conclusions: Our findings indicate that omitting extended post-injection instructions does not negatively impact patient satisfaction or complication rates. Given the toxin’s rapid internalization and localized effect, extended behavioral restrictions may be redundant. However, the absence of a control group and lack of statistical analyses limit the strength of these conclusions. In addition, this is a short-term study. Future prospective, randomized trials are needed to develop evidence-based post-care protocols to optimize esthetic outcomes, patient safety, and long-term efficacy.

## 1. Introduction

Botulinum toxin (BoNT), produced by the anaerobic bacterium *Clostridium botulinum*, exists in seven serological types, with BoNT type A [1] being the most widely used for medical purposes. The toxin acts through a multi-step process—recognition, binding, and catalysis—ultimately inhibiting neurotransmitter release by cleaving SNAP-25 (Synaptosomal-Associated Protein of 25 kDa) [2], VAMP (Vesicle-Associated Membrane Protein), and syntaxin, thereby disrupting synaptic vesicle fusion and inducing paralysis in peripheral neurons [3,4].

As a neuromodulator, BoNT type A effectively reduces dynamic facial wrinkles, including forehead [5], glabellar [6], and crow’s feet lines, and is used for lifting eyebrows and increase eye widening either alone or in combination with other rejuvenation techniques.

Numerous studies have demonstrated the significant efficacy of BoNT in reducing wrinkles, underscoring the importance of individualized treatment plans for achieving the best results. Despite its proven safety profile, with rare and mild complications often related to injection technique, the post-treatment instructions normally given to patients for the following 4 h are as follows:-Avoiding facial expressions;-Staying upright;-Refraining from physical activity;-Avoiding direct sun;-Avoiding heat sources [7].

Those recommendations are largely based on anecdotal evidence rather than robust clinical research [8].

By comparing the clinical results and patient experiences in this group, we seek to determine whether strict compliance with traditional post-injection protocols is essential for achieving optimal treatment outcomes.

Finally, we must remember that Botulinum toxin A is derived from a potent neurotoxin, considered one of the most lethal substances known, with an estimated lethal dose of approximately one nanogram per kilogram of body weight. Its medical use dates back nearly two centuries, when the German physician Justinus Andreas Christian Kerner first recognized its effects on skeletal muscle function and parasympathetic nervous activity. Since then, botulinum toxin has been developed into a therapeutic tool, widely used in both medical and cosmetic treatments, including the management of muscular disorders, chronic migraines, and hyperhidrosis. Its precise mechanism of action involves blocking the release of acetylcholine at the neuromuscular junction, leading to muscle paralysis [9]. However, it is not yet free from complications.

The objectives of this study are the following:-To summarize the scientifically proven indications for botulinum toxin treatment to date.-To evaluate the necessity of conventional post-treatment care instructions following botulinum toxin injections by assessing patient satisfaction in those who did not adhere to these guidelines beyond the first 10 min after injection.-To compare the prevalence of post-treatment complications, particularly upper eyelid ptosis, in patients following the 10 min protocol versus those adhering to conventional post-injection guidelines, using data from the existing literature.

In this context, we conducted a multicenter retrospective study involving over 5000 patients to assess whether strict adherence to traditional post-injection guidelines is necessary. Through analysis of clinical outcomes and patient-reported satisfaction, our work aims to provide evidence regarding the safety and validity of a simplified 10 min post-treatment protocol.

## 2. Results

A cohort of a total of 5014 patients was included in the final analysis. They were primarily female and aged between 18 and 69 years, and underwent botulinum toxin treatment in the upper third of the face. The study aimed to assess whether strict adherence to post-treatment guidelines affected the risk of developing upper eyelid ptosis.

Patients were taught the post-treatment procedures and rules to respect in the immediate 10 min after the injection and were supervised until the end of the treatment session.

Two weeks after the treatment, the patients were clinically re-evaluated and were asked for their personal opinion about the results obtained through a questionnaire.

### 2.1. Patient Demographics and Baseline Characteristics

The demographic distribution and baseline characteristics of the patients are summarized in Table 1.

### 2.2. Patient Satisfaction

Patient satisfaction was assessed one month after the treatment by fulfilling a standardized questionnaire. Not all 5014 patients responded to the questionnaire, and a total of 2010 questionnaires were finally collected. Satisfaction levels are summarized in Table 2.

As shown, the majority of patients (90%) reported being either “Very Satisfied” or “Satisfied” with their treatment outcomes.

### 2.3. Side Effects

The incidence of side effects was carefully monitored and examined during the one-month after-treatment follow-up. Of the 5014 patients treated, 4000 (79.8%) attended the scheduled clinical evaluation. The most commonly reported adverse effects are summarized in Table 3.

The overall incidence of side effects was 6.00%, with no severe adverse reactions reported.

Notably, no cases of upper eyelid ptosis were observed among the 4000 patients evaluated at follow-up, underscoring the safety of the 10 min post-treatment protocol and reinforcing the primary endpoint of the study.

## 3. Discussion

After having searched the literature, to the authors’ knowledge, there are no clinical trials assessing the impact of the common post-treatment recommendations (such as remaining upright and avoiding certain activities) after botulinum toxin type A (BoNT-A) injections, nor their efficacy and durability supported by scientific evidence. However, evidence has been found as follows.

Recent emerging evidence suggests that muscle activity immediately following botulinum toxin (BoNT-A) injection may potentially improve treatment outcomes. However, these findings are still in the early stages and require further validation, especially within the context of esthetic applications. This area presents an opportunity for future clinical trials to explore in greater depth, aiming to understand how post-injection muscle activity may influence the diffusion and binding of BoNT-A, thereby refining treatment protocols and enhancing clinical efficacy [10. Nonetheless, in our study design, we chose not to assess long-term efficacy related to post-injection muscle activity, as it is subject to substantial interindividual variability (differences in metabolism, muscle mass, genetic factors, lifestyle, and prior exposure), which would have introduced significant confounding. Instead, we focused on short-term outcomes such as adverse events and patient satisfaction, which are more reliably assessed within a standardized 2-week follow-up window.

Up-to-date studies have also questioned the common practice of cooling the injection site to minimize bruising and swelling. At the same time, initial research suggests that cooling may impede BoNT translocation, which could potentially reduce the treatment’s efficacy and negatively affect toxin distribution. Moreover, the toxins are zinc-dependent proteases, and the supplemental intake of zinc could enhance their effect. Moreover, given the rapid internalization of BoNT-A, as shown in studies where the toxin is internalized within minutes [10], the need for extended post-treatment restrictions is questionable. Research demonstrates that binding and internalization at the neuromuscular junction occur quickly, suggesting that any complications would most likely manifest within a short timeframe post-injection. As such, extended post-treatment restrictions, which are commonly advised, may not provide additional benefits.

Furthermore, studies have shown variability in the onset and action among different BoNT/A subtypes. For instance, Pellett et al. (2015) showed that each BoNT/A subtype has a distinct profile that can influence clinical outcomes [11]. This finding is crucial for tailoring treatment based on specific therapeutic or cosmetic goals, especially when considering the desired speed of onset and duration of effect. Such understanding reinforces the importance of selecting the appropriate BoNT/A subtype for each individual patient, but it also highlights that post-treatment care might need to be adjusted based on the specific toxin used.

Finally, it has been confirmed that Botulinum toxin (BoNT) does not possess an intrinsic ability to “spread” in the way it is often described. At the recommended doses, BoNT acts locally and remains confined to the injection area. Diffusion—rather than spread—is the correct term, and it is dose-dependent: a higher dose allows the toxin to gradually diffuse further from the injection site, expanding the area of effect. This principle has been demonstrated in numerous clinical studies and can be advantageous for certain treatments. Any variations observed in diffusion profiles between different BoNT products can typically be addressed by adjusting the dose. However, serious complications, such as ptosis of the eyelid, may arise if the injector lacks the necessary expertise or experience to administer the toxin correctly [12].

Ultimately, it is essential to remember that the systemic toxicity of botulinum toxin in humans is estimated at approximately 1 ng per kilogram of body weight [13].

All the above-mentioned aspects are summarized in Table 4.

To respond to the second aim of the study (as shown in Table 3), the distribution of patient satisfaction levels following botulinum toxin (BoNT-A) treatment is shown. Among the 2010 patients of the 5014 included in the study who responded to the questionnaire, a substantial majority (90%) reported a positive experience, with 50% (n = 1005) being “Very Satisfied” and 40% (n = 804) “Satisfied” with their treatment outcomes. A smaller proportion of patients (6%, n = 120) expressed a neutral stance, while dissatisfaction remained minimal, with only 3% (n = 60) reporting being “Very Dissatisfied” and 1% (n = 21) “Dissatisfied.” These findings indicate a high overall satisfaction rate and suggest that adherence to post-treatment care beyond the initial 10 min did not significantly influence patient-reported outcomes.

As for the last objective of the study, the general incidence of adverse events (AEs) for Botulinum Toxin A (BoNT-A) in a review by Z. Jia et al. reported a statistically significant increase in AEs in the BoNT-A group compared to the placebo group, with a relative risk (RR) of 1.24 (95% CI 1.07–1.43, *p* = 0.003). Specific adverse events that were more frequent included headaches, eyelid ptosis, heavy eyelids, and injection site hematoma, particularly in the treatment of glabellar lines (GL). Despite this, most of these events were mild to moderate in severity [13].

In another study, the overall complication rate after BoNT-A injections was 16%, with common issues including headache, local skin reactions, and facial neuromuscular symptoms, primarily mild and transient [14].

According to Sethi et al., the incidence of adverse effects related to Botulinum toxin A (BoNT-A) injections for facial rejuvenation in the upper third of the face, including glabellar lines and crow’s feet, showed a significant but mild-to-moderate safety profile [15].

In our study, instead, in a total number of 5014 patients we reported mild bruising in 120 cases (3.00%), temporary headache in 72 cases (1.80%), eyebrow ptosis in 16 cases (0.40%), dry eyes in 12 cases (0.30%), and other adverse effects (e.g., muscle weakness, flu-like symptoms) in 20 cases (0.50%), as shown in Table 3.

This gave us a total of 240 adverse events with an overall incidence of adverse effects in the study of 6.00%.

We find it appropriate to mention, in conclusion, that among the most frequently minor esthetic side effects following upper facial BoNT-A treatment is the so-called “quizzical” or “cocked” eyebrow, resulting from uneven frontalis muscle activity. Although easily manageable with minor touch-up injections and not clinically significant, its high frequency in clinical practice warrants further attention. Due to the retrospective nature of this study, this specific outcome was not systematically recorded, and thus not included in the adverse events analysis. Future prospective studies should consider including it among standardized outcome measures.

To conclude, the results of this study, showing no significant increase in adverse effects and high patient satisfaction despite the absence of stringent post-treatment care, support the need to re-evaluate traditional post-injection recommendations. Injection-site reactions like erythema, edema, and ecchymosis remain the most common side effects. Therefore, precautions such as avoiding nonsteroidal anti-inflammatory drugs (NSAIDs) and ensuring sterile injection techniques remain vital for minimizing these risks. Additionally, advising patients to avoid agents that impair clotting several days before the procedure can assist mitigate the risk of common adverse effects, such as bruising. Ensuring proper sterilization and advising patients to remove all makeup before treatment are also essential steps in reducing the risk of infection.

Considering these findings, there is a clear need for well-designed, large case studies to establish evidence-based guidelines for post-treatment care following BoNT-A injections. Future research should focus on optimizing treatment efficacy, reducing adverse effects, and improving patient outcomes. This study provides a foundation for further exploration into the necessity and effectiveness of common post-injection care practices, ultimately promoting a more evidence-based approach to BoNT-A treatments.

## 4. Conclusions

This study challenges several long-held beliefs about post-botulinum toxin care, suggesting that many traditional recommendations may not be necessary. As BoNT-A is rapidly internalized, the timeframe for potential complications is brief, and extended restrictions may be unnecessary. It is important to note that our study did not assess the impact of the simplified 10 min protocol on long-term efficacy, such as the duration of BoNT-A effects. Therefore, future research should explore this aspect further to refine post-treatment guidelines and optimize both safety and patient outcomes.

## 5. Materials and Methods

This retrospective, multicentric study was conducted across six major centers in Italy—Milan, Modena, Florence, Rome, Naples, and Catania—to ensure comprehensive geographical coverage of the country’s regions. The study spanned from January 2015 to December 2020, providing a broad and representative sample of patients treated with botulinum toxin type A (BoNT-A). Data were taken from the databases of patients from every single center involved in the study.

The data were analyzed using descriptive statistics through Microsoft Excel (version 365, Microsoft Corporation, Radmond, WA, USA), reporting absolute and relative frequencies of adverse events and satisfaction levels. No inferential statistical correlations were applied, as this was an observational descriptive study.

The primary outcome was the incidence of complications, above all upper eyelid ptosis, assessed through clinical evaluation. The second outcome was patient-reported satisfaction evaluated through a questionnaire.

### 5.1. Inclusion Criteria Were

Age between 18 and 69 years;Signs of dynamic or static facial wrinkles in the upper third of the face;Patient’s desire to rejuvenate their appearance.

### 5.2. No Treated Patients Were Those Who Had

Contraindications to botulinum toxin type A (BoNT-A), such as keloidal scarring, body dysmorphic disorder, pregnancy, and breast feeding [16];Previous adverse reactions to BoNT-A;Underlying neurological disorders that could interfere with treatment outcomes.

All patients signed an informed consent form outlining the treatment, its purpose, expected outcomes, and potential risks and benefits. This ensured patients were fully informed about the procedure and its implications.

The data collected for each patient included demographic information (age, sex, and medical history), as well as any adverse effects experienced post-treatment. Adverse effects were categorized into common, rare, and severe complications, including ptosis, asymmetry, or unattended diffusion of BoNT-A. Additionally, data on patient adherence to postoperative instructions were collected, though specific instructions varied across centers.

Descriptive statistics were used to summarize patient demographics and the incidence of adverse effects. To assess the efficacy of non-specific postoperative care instructions, complication rates observed in this study were compared to complication rates reported in the existing literature.

The analysis focused on determining whether the rates of adverse effects in patients who received non-specific postoperative instructions were comparable to those reported in the literature, where stricter postoperative guidelines were followed.

### 5.3. Injection Technique

The injection protocol followed general anatomical and clinical principles for esthetic treatment of the upper third of the face [17]:

Forehead wrinkles: Injections were administered subcutaneously in two rows, tailored to the type of muscle contraction and the patient’s age, all placed in the “safe area” approximately 2 cm above the eyebrow to avoid brow ptosis.

Glabellar wrinkles: Botulinum toxin was injected at the periosteum at the origin of the corrugator superficilii muscle and subcutaneously at its insertion, with a standardized dose at each site. An additional standardized dose was injected into the center of the glabellar area to block the procerus muscle.

Lateral Periorbital Lines: Injections were administered subcutaneously at three sites around the lateral canthus, with a standardized dose evenly distributed at each site.

The total dose administered per patient was adjusted according to clinical judgment and patient needs.

### 5.4. Post-Treatment Protocol

Patients were observed in the clinic for 10 min to ensure that no immediate adverse reactions occurred. No specific post-treatment instructions or restrictions were given. Patients were free to resume their normal activities immediately after leaving the clinic. Patients were asked to return for a follow-up visit two weeks after treatment and were asked to fulfill a satisfaction questionnaire based on a 5-point Likert scale, ranging from “Very Satisfied” to “Very Dissatisfied,” (as reported in Table 3).

Adverse effects were collected retrospectively based on clinical documentation and included all commonly reported reactions such as bruising, headache, dry eyes, and upper eyelid ptosis. Ptosis was clinically assessed during the two-week follow-up visit by experienced physicians. No allergic reactions were reported in the analyzed cohort.

## Figures and Tables

**Table 1 toxins-17-00372-t001:** Baseline characteristics of the study population.

Variable	Mean ± SD/N (%)
Age (years)	47.3 ± 9.2
Female: 4513 (90.0%)	
Male: 501 (10.0%)	

**Table 2 toxins-17-00372-t002:** Patient satisfaction level.

Satisfaction Level	N (%)
Very Satisfied	1005 (50%)
Satisfied	804 (40%)
Neutral	120 (6%)
Dissatisfied	60 (3%)
Very Dissatisfied	21 (1%)

**Table 3 toxins-17-00372-t003:** Incidence of side effects.

Side Effect	N (%)
Mild bruising at injection site	120 (3.00%)
Temporary headache	72 (1.80%)
Eyebrow ptosis	16 (0.40%)
Dry eyes	12 (0.30%)
Others (e.g., muscle weakness, flu-like symptoms)	20 (0.50%)

**Table 4 toxins-17-00372-t004:** Scientific evidence about the post-treatment common knowledge so far.

Aspect	Current Practice	Evidence Status	Recommendations
Post-treatment instructions and restrictions	Widely used	Lacks robust evidence of efficacy or adverse event reduction	Further research needed to establish evidence-based guidelines
Muscle activity post-injection	Not typically advised	Preliminary evidence suggests benefits	More studies needed to confirm relevance to esthetic treatments
Temperature influence on BoNT translocation	Cooling the area is common	Initial evidence suggests cooling may reduce efficacy	Cooling should be reconsidered; further investigation required
Migration of the toxin	No touch nor massage	The toxin acts locally and remains confined to the injection area	Adjust the dose

## Data Availability

The data supporting the findings of this study are held by the individual surgeons at the respective centers that participated in the study.

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
