# Peer review of "Are Post-Care Recommendations Following Upper-Face Botulinum Toxin Treatment Scientifically Necessary? A Retrospective Study Based on 5000 Patients"

_toxins, 2025, doi:10.3390/toxins17080372_

Round 1
Reviewer 1 Report
Comments and Suggestions for Authors
I also agree with the concern raised in this paper regarding how clinicians often impose post-treatment lifestyle restrictions on patients without solid evidence. In that sense, I appreciate the effort and value of conducting this meaningful study.
I would like to offer a few comments:
1. Clarification of Treatment Area in the Title
This study focuses solely on upper facial botulinum toxin treatments—namely the forehead, glabella, and crow’s feet. Therefore, the findings should not be generalized to all botulinum toxin procedures. It would be more accurate to include "upper face botulinum toxin treatment" in the title.
2. Clarification and Accuracy of Referenced Guidelines
The manuscript references guidelines in citations [7] and [8], but some discrepancies with the actual content of those references are noted.
- Reference [7] states:
"Patients may resume normal activities after 4 hours. There is no need to move, or avoid moving, the affected muscles. Some practitioners recommend contracting the treated muscles to facilitate uptake of the product, but this is not necessary." - Reference [8] notes:
"Patients should refrain from touching the treated area for 3–4 hours after administration. Following treatment, patients should exercise the treated muscles by tensing them and relaxing them. This helps to work the botulinum toxin into the muscle. Sunscreen application is also recommended post-injection to reduce the risk of denaturation or degradation. Patients should avoid lying down for 4 hours post-injection and refrain from strenuous exercise."
As seen above, these sources do not recommend restricting normal daily activities for 24 hours. It may be helpful to revise the description of existing post-treatment recommendations so they are not overstated.
3. Purpose of Traditional Post-Treatment Guidelines
Traditional post-treatment guidelines were not only implemented to reduce adverse events but also to optimize efficacy. For example, exercising the treated muscle was advised to increase toxin uptake. While reducing or eliminating this step may not increase adverse events, it may potentially reduce the magnitude or longevity of clinical effects. Likewise, sunscreen use post-injection is intended to prevent product degradation—not to prevent adverse effects.
Therefore, concluding that a modified 10-minute rule is superior just because adverse events were not increased may be premature. The 2-week follow-up used in this study does not capture long-term efficacy or duration of effect. This limitation should be acknowledged in the discussion.
4. Formatting of Scientific Terminology
Clostridium botulinum should be italicized throughout the manuscript.
5. Consistency in Terminology
The terms BTX-A and BoNT-A seem to be used interchangeably. It would be preferable to choose one and use it consistently throughout the manuscript.
6. Redundancy
Some points in the manuscript are repeated unnecessarily. The text could be streamlined for clarity and conciseness without losing any important information.
7. Introduction Content
The basic introduction to botulinum toxin and its historical background may not be necessary in this context and could be omitted to enhance focus and flow.
8. Minor but Common Adverse Events
One of the most common adverse effects of upper facial botulinum toxin treatment is the so-called “quizzical” or “cocked” eyebrow. While minor and manageable, its high frequency warrants attention. Was data on this particular adverse event collected?
Author Response
Dear Reviewer,
We would like to thank you for your review of our manuscript. Your comments have been helpful in improving the clarity, accuracy, and scientific rigor of our work. Please find below our responses to each of your points:
Comment 1:
- Clarification of Treatment Area in the Title
This study focuses solely on upper facial botulinum toxin treatments—namely the forehead, glabella, and crow’s feet. Therefore, the findings should not be generalized to all botulinum toxin procedures. It would be more accurate to include "upper face botulinum toxin treatment" in the title.
Response1:
We agree with your observation. Accordingly, we have revised the manuscript title to:
"Are Post-Care Recommendations Following upper-face botulinum toxin Treatment Scientifically Necessary? A Retrospective Study Based on 5000 Patients"
Comment 2:
- Clarification and Accuracy of Referenced Guidelines
The manuscript references guidelines in citations [7] and [8], but some discrepancies with the actual content of those references are noted.
- Reference [7] states:
"Patients may resume normal activities after 4 hours. There is no need to move, or avoid moving, the affected muscles. Some practitioners recommend contracting the treated muscles to facilitate uptake of the product, but this is not necessary." - Reference [8] notes:
"Patients should refrain from touching the treated area for 3–4 hours after administration. Following treatment, patients should exercise the treated muscles by tensing them and relaxing them. This helps to work the botulinum toxin into the muscle. Sunscreen application is also recommended post-injection to reduce the risk of denaturation or degradation. Patients should avoid lying down for 4 hours post-injection and refrain from strenuous exercise."
As seen above, these sources do not recommend restricting normal daily activities for 24 hours. It may be helpful to revise the description of existing post-treatment recommendations so they are not overstated.
Response 2:
Thank you for highlighting this. We acknowledge that our previous phrasing may have overstated the rigidity of traditional post-treatment recommendations. We have revised the relevant section in the discussion to reflect the nuanced nature of existing guidelines, indicating that many recommendations are based on habit or tradition rather than robust evidence.
Comment 3:
- Purpose of Traditional Post-Treatment Guidelines
Traditional post-treatment guidelines were not only implemented to reduce adverse events but also to optimize efficacy. For example, exercising the treated muscle was advised to increase toxin uptake. While reducing or eliminating this step may not increase adverse events, it may potentially reduce the magnitude or longevity of clinical effects. Likewise, sunscreen use post-injection is intended to prevent product degradation—not to prevent adverse effects.
Therefore, concluding that a modified 10-minute rule is superior just because adverse events were not increased may be premature. The 2-week follow-up used in this study does not capture long-term efficacy or duration of effect. This limitation should be acknowledged in the discussion.
Response 3:
We thank the reviewer for this observation.
However, we respectfully note that current evidence and international guidelines are not unanimous regarding the benefit of post-injection muscle activity (such as actively contracting the treated muscles) to enhance toxin uptake and other common beliefs suggest limiting excessive physical movement and strenuous activity in the hours following injection to reduce the risk of undesired diffusion, which could compromise precision and increase adverse events such as asymmetry or ptosis.
Furthermore, as correctly mentioned, our study primarily aimed to assess safety (adverse event rate) and patient satisfaction in the context of a simplified 10-minute post-treatment protocol. In this regard, we would like to clarify that the follow-up at 2 weeks was deliberately chosen, as the clinical peak of botulinum toxin activity—as well as the manifestation of the majority of complications—typically occurs within 7–10 days. Thus, we considered this follow-up window sufficient to detect any clinically relevant differences in outcome between the traditional protocol and the modified approach.
As for the duration of treatment effect, while we agree that this is an important endpoint, we deliberately chose not to include it among our study outcomes due to the high degree of interindividual variability—including differences in metabolism, muscle mass, genetic factors, lifestyle, and prior exposure—which would have introduced significant confounding. These factors would require a separate, well-controlled prospective study with standardized toxin dosing and patient profiling.
We added this to the text so that il can be better understood by the readers too.
Comment 4:
- Formatting of Scientific Terminology
Clostridium botulinum should be italicized throughout the manuscript.
Response 4:
We thank you for pointing this out. We corrected it.
Comment 5:
- Consistency in Terminology
The terms BTX-A and BoNT-A seem to be used interchangeably. It would be preferable to choose one and use it consistently throughout the manuscript.
Response 5:
You have corrected by using BoNT-A throughout the entire work in accordance with the more commonly accepted scientific usage.
Comment 6:
- Redundancy
Some points in the manuscript are repeated unnecessarily. The text could be streamlined for clarity and conciseness without losing any important information.
Response 6:
We thank you for the observation regarding possible redundancies in the manuscript.
After careful internal revision, all co-authors have thoroughly reviewed the full text and collectively agreed that the current structure and content effectively support the scientific rationale of the study. While we acknowledge that some concepts are reiterated, this was done intentionally to emphasize key messages that are central to our hypothesis and to guide the reader through the logical progression toward our defined outcomes.
In particular, given that our work challenges a long-standing and widely accepted clinical practice, we felt it was necessary to reinforce certain arguments to ensure that both the context and the justification for our conclusions are conveyed clearly and convincingly.
Nonetheless, we remain fully open to making specific adjustments if the editorial team deems it appropriate, and we are grateful for the opportunity to clarify our reasoning.
Comment 7:
- Introduction Content
The basic introduction to botulinum toxin and its historical background may not be necessary in this context and could be omitted to enhance focus and flow.
Response 7:
We appreciate the reviewer’s suggestion regarding the historical introduction.
However, we believe that this concise overview—which is limited to just a few lines—allows the reader to briefly revisit the development and the basic mechanism of action of botulinum toxin which, we believe, are useful for contextualising the clinical implications of our work.
In addition, this may be helpful for non-specialist readers or for those who are approaching the topic from a different medical background.
That said, we have reviewed the section again and we have shorten it again.
We are open to further streamlining if deemed necessary by the editorial team.
Comment 8:
- Minor but Common Adverse Events
One of the most common adverse effects of upper facial botulinum toxin treatment is the so-called “quizzical” or “cocked” eyebrow. While minor and manageable, its high frequency warrants attention. Was data on this particular adverse event collected?
Response 8:
We thank you for bringing attention to the “quizzical” or “cocked” eyebrow effect, which is indeed one of the most frequent minor aesthetic complications following upper facial BoNT-A treatment.
Due to the retrospective and multicentric nature of our study, adverse events were recorded only when explicitly documented in the medical records or flagged during follow-up visits. Unfortunately, this specific outcome—being a minor, non-pathological, and easily correctable asymmetry—was not systematically collected across all centers and was therefore not included in the structured adverse event data presented.
We agree that this effect deserves further attention in prospective studies, and we have added a sentence in the discussion to acknowledge its clinical relevance and high frequency in daily practice.
We are grateful for your constructive feedback, which has significantly improved the manuscript. Please let us know if any further revisions are needed.
Sincerely,
Reviewer 2 Report
Comments and Suggestions for Authors
Dear Authors,
The subject of the manuscript is interesting and the number of patients taken into the study is impressive
The abstract includes a figure that is not mentioned in the text, is not numbered and I think it is not appropriate to find it in the abstract - line 26
The introduction is appropriate to the subject, but it is very fragmented and I believe it should have a sentence at the end mentioning the current study.
In the material and method - the statistical program used and the way of assessing the statistical results
Are not mentioned the patient satisfaction scale used is not mentioned the statistical correlations used
Rows 83-87 from the results
I think it would be appropriate to mention in the material and method what other aspects were taken into consideration - allergic reactions,.. how was the eyelid ptosis assessed
I believe that the assessment of results must be done from several points of view - injection technique, injection points, allergies,...
Discussions should consider comparisons between the results of the present study and those of the same kind reported in the literature
Best regards,
Author Response
Dear Reviewer,
We thank you for your comments and valuable suggestions. We are grateful for your positive feedback on the relevance of the subject and the size of our study population. Below are our detailed responses to each of your observations
The subject of the manuscript is interesting and the number of patients taken into the study is impressive.
We would like to sincerely thank you for the kind words and encouraging feedback regarding the relevance of our topic and the size of our patient cohort. We truly appreciate your recognition of the study's scope and scientific potential, and we are grateful for the time and attention you have dedicated to improving our manuscript.
The abstract includes a figure that is not mentioned in the text, is not numbered and I think it is not appropriate to find it in the abstract - line 26
The image mentioned below the abstract is a graphical abstract that complies with MDPI’s GA (Graphical Abstract) guidelines. As such, it is neither numbered nor referenced within the main text, but positioned according to the submission platform's requirements and formatting instructions. However, if necessary for the editorial team, we could relocate the image to another section of the manuscript.
The introduction is appropriate to the subject, but it is very fragmented and I believe it should have a sentence at the end mentioning the current study.
We have revised the introduction to improve its flow and coherence, and we have added a closing sentence that states the aim and rationale of the current study, thereby linking the background to our research objectives.
In the material and method - the statistical program used and the way of assessing the statistical results are not mentioned
The data were analyzed using descriptive statistics through Microsoft Excel, reporting absolute and relative frequencies of adverse events and satisfaction levels. No inferential statistical correlations were applied, as this was an observational descriptive study. We added that to the study.
the patient satisfaction scale used is not mentioned the statistical correlations used
Patient satisfaction was assessed using a 5-point Likert scale, ranging from "Very Satisfied" to "Very Dissatisfied," as reported in Table 3. We added that to the text.
I think it would be appropriate to mention in the material and method what other aspects were taken into consideration - allergic reactions,.. how was the eyelid ptosis assessed
Adverse effects were collected retrospectively based on clinical documentation and included all commonly reported reactions such as bruising, headache, dry eyes, and upper eyelid ptosis. Ptosis was clinically assessed during the two-week follow-up visit by experienced physicians. No allergic reactions were reported in the analyzed cohort. All this information was already presented in the Results section. However, following your recommendation, we have now added a concise description of these aspects in the Materials and Methods section as well, to improve clarity and methodological transparency. We remain open to further discussing this point should the editorial team or reviewers deem it necessary.
I believe that the assessment of results must be done from several points of view - injection technique, injection points, allergies,...
We confirm that all patients were treated following a standardized injection technique, with consistent injection points and dosages as outlined in the Injection Technique section. This consistency allowed for uniform analysis across all six centers.
Discussions should consider comparisons between the results of the present study and those of the same kind reported in the literature
We thank you for this thoughtful suggestion. In fact, this is exactly what we aimed to do in the Discussion section. Throughout the text, we compared our findings with those reported in already-existing studies, highlighting both consistencies and divergences where relevant. While some characteristics mentioned in the literature—such as toxin diffusion, muscle activity post-injection, and the rationale behind traditional care instructions—were not primary endpoints in our study, they were nonetheless discussed in relation to our results when relevant.
Our intent was to contextualize our observations within the broader scientific landscape, acknowledging similarities with existing data and contributing additional insight from a large real-world cohort. We hope this clarifies that literature comparison was not only considered but actively integrated into our discussion.
We are grateful for your constructive feedback, which has significantly improved the manuscript. Please let us know if any further revisions are needed.
Sincerely,
Round 2
Reviewer 1 Report
Comments and Suggestions for Authors
Thank you for your thoughtful and active revisions in response to my previous comments.
I would like to respectfully raise just two additional points:
1.
I fully understand and agree with the main conclusion of this study—that the simplified 10-minute post-treatment protocol did not increase adverse events or affect patient satisfaction at the 2-week follow-up, when compared to traditional post-injection recommendations.
I also appreciate your clear explanation as to why long-term efficacy (such as duration of effect) was not assessed in this study.
However, I would suggest explicitly stating in the manuscript that the impact of the 10-minute protocol on long-term efficacy—such as duration of effect—has not been evaluated.
This clarification is important, as the current title may unintentionally imply that existing post-treatment guidelines are universally unnecessary.
I recommend narrowing the scope of your conclusions to specify that the simplified protocol appears unnecessary in terms of short-term safety and satisfaction only, as evaluated in this study.
Avoiding broader generalizations will enhance the clarity and scientific rigor of your message.
2.
In line 34, the terminology should be standardized. The correct initial term is Botulinum toxin (BoNT), and subsequently, Botulinum toxin type A should be abbreviated as BoNT-A after the first mention.
Author Response
We sincerely thank the reviewer for the comments and for the constructive suggestions, which have contributed to the improvement of our manuscript. Please find below our responses:
1.
I fully understand and agree with the main conclusion of this study—that the simplified 10-minute post-treatment protocol did not increase adverse events or affect patient satisfaction at the 2-week follow-up, when compared to traditional post-injection recommendations.
I also appreciate your clear explanation as to why long-term efficacy (such as duration of effect) was not assessed in this study.
However, I would suggest explicitly stating in the manuscript that the impact of the 10-minute protocol on long-term efficacy—such as duration of effect—has not been evaluated.
This clarification is important, as the current title may unintentionally imply that existing post-treatment guidelines are universally unnecessary.
I recommend narrowing the scope of your conclusions to specify that the simplified protocol appears unnecessary in terms of short-term safety and satisfaction only, as evaluated in this study.
Avoiding broader generalizations will enhance the clarity and scientific rigor of your message.
We agree with your observation.
In response to your suggestion:
We have revised both the abstract and the conclusion sections to clearly state that our findings relate specifically to short-term outcomes.
We believe this improves the precision and scientific rigor of our message, and we are grateful for your recommendation.
2.
In line 34, the terminology should be standardized. The correct initial term is Botulinum toxin (BoNT), and subsequently, Botulinum toxin type A should be abbreviated as BoNT-A after the first mention.
Thank you for highlighting this. As suggested:
We have revised the manuscript to ensure correct and consistent terminology throughout.
We truly appreciate your careful review and valuable insights.
With kind regards,
On behalf of all co-authors
Reviewer 2 Report
Comments and Suggestions for Authors
Dear Authors,
The authors improved the content and respected the recommendations
Best regards,
Author Response
Dear Reviewer,
We sincerely thank you for your kind feedback and for recognizing the improvements made to the manuscript. We greatly appreciated your thoughtful recommendations throughout the review process and are pleased that the revised version meets your expectations.
With kind regards,
All the authors